# A Feature-Extraction-Based Adaptive Refinement Method for Solving the Reynolds Equation in Piston–Cylinder System

**Jiashu Yang [1], Bingquan Zuo [2],\*, Huixin Luo [2] and Weikang Xie [1]**

[1]  Key Laboratory of Metallurgical Equipment and Control Technology Ministry of Education,
    Wuhan University of Science and Technology, Wuhan 430081, China
[2]  Hubei Key Laboratory of Mechanical Transmission and Manufacturing Engineering,
    Wuhan University of Science and Technology, Wuhan 430081, China
\*  Correspondence: zuobq@wust.edu.cn or zuobingquan@gmail.com

**Abstract:** A fast local refinement algorithm based on feature extraction is developed. In the mesh-based Reynolds equation solutions, two refinement features based on the physical parameters of fluid lubrication are firstly defined, namely, pressure value feature and pressure gradient feature. Then, a fast adaptive strategy different from the traditional methods based on residuals or recovery errors is constructed according to the features, which are expected to determine the element needed to be refined. Considering the update requirement of the feature parameters, an adaptive update strategy for feature parameters is also developed. Finally, the feasibility of the scheme is verified on a single-cylinder gasoline engine. Results show that the current algorithm can effectively reduce the computational scale while ensuring the computational accuracy of the mesh-based model, compared with the traditional global and local refinement strategy.

**Keywords:** feature extraction; Reynolds equation; mesh-based model; adaptive refinement

## 1. Introduction

The state of lubrication has a great influence on the performance of the whole machine, and it accounts for the major sources of mechanical noise and vibration [1], especially in the piston–cylinder system. The piston–cylinder system is a less stable rigid system, consisting of the piston inertial dynamics model and the lubrication oil film model. The integral method with a variable stepsize at each integration step adaptive time-setup size is generally used to ensure the solution accuracy. In a simulation of the piston–cylinder system, the solving process of the lubrication film pressure will be repeated for millions of times in the solution of a whole workflow [2]. Obviously, a method that performs well in terms of solution speed while maintaining high accuracy can significantly reduce computation time.

Stable and efficient numerical solutions to fluid problems have always been a focus, and plenty of methods have been developed, which can be divided into mesh-based methods and mesh-free methods [3] based on the way of the computational domain discretization. These include finite element method [4] (FEM), finite difference method [5] (FDM), finite volume method [6] (FVM), and isogeometric analysis method [7] (IGA), which belong to the former. These methods discretize and solve the computational domain and governing equations based on the mesh as the basic unit [8]. Usually, they have high algorithmic stability when dealing with fluid problems with fixed boundary or small deformation, but due to the strict mesh deformation conditions, these methods are not effective or are even unapplicable when dealing with large deformations or free flows [9]. Mesh-free methods have been introduced into fluid solutions. These methods use points or particles as the basic unit for discretization, instead of mesh, and can effectively handle problems such as large deformations [10], free flows [11], and multiphase flows [12]. One typical method is the smoothed particle hydrodynamics [13,14] (SPH) method, which uses

a set of freely movable particles to discretize the fluid and related control equations. This flexible handling method can directly satisfy mass conservation in flow and is applicable to fluid problems in complex spaces [15]. However, it should be pointed out that the position and distribution of points or particles used in the method will seriously affect the solution accuracy, and there are certain limitations in model adaptation. Specifically, in the simulation analysis of piston–cylinder friction pairs studied in this paper [16], the lubrication area is usually described by the Reynolds equation (a second-order partial differential equation) with a simple form of boundary, which is very suitable for mesh-based methods to solve, and related works include FDM [17], FEM [18], FVM [19], IGA [20], etc.

Generally, the mesh scale directly determines the model's calculation time and accuracy. In order to improve efficiency, the adaptive subdivision method is widely used in classic mesh-based methods [21], which has tried to build a "proper" mesh [22] to reconcile the conflict of the decrease in solving efficiency and the increase in accuracy with the increasing of mesh size. These methods mainly use residual posterior error [23] and recovery posterior errors [24] for error estimation. As the core parts of the adaptive refinement process, the selection of the super convergence points has great influence on the calculation of the residual posterior error, making it very inefficient. While the recovery posterior error depends heavily on the equation's form, the variable coefficients of the Reynolds equation making it cannot be directly applicable [25]. It should be noted that it is quite complicated for FVM to handle the control volumes and the boundary conditions between them in refinement.

In this paper, IGA, known as "the next generation of FEM" because of its efficiency, is used to solve the Reynolds equation, and PHT splines that support the local subdivision are used to mesh. Based on this, a feature-extraction-based adaptive refinement strategy is proposed in this paper. As the target is solving the distribution of lubrication film pressure, the physical features of the film were taken into consideration for the refinement strategy, such as the pressure value and its variation. It is obvious that the two parameters indicate the distribution trend of the lubrication film pressure. Thus, the refinement strategy is proposed in this article based on the following empirical assumption: the calculation results of the regions with large pressure value or rapid variation in pressure value have greater influence on the accuracy of the calculation of the mechanical parameters of the friction sub. Furthermore, the calculation accuracy of the regions also impacts the calculation accuracy of the whole system greatly. Thus, it is crucial that these regions are be refined. Particularly, for the JFO boundary condition [26], the calculation of cavity regions is absolutely different from the fully lubricated areas. The cavity feature is defined to mark the elements in cavity regions particularly. In other words, the refinement method can make the solving of the Reynolds equation with JFO boundary conditions easier. All the definitions of abbreviations in this paper can be find in Table 1.

**Table 1.** Abbreviations and definitions.

| Abbreviations | Definitions |
|:---:|:---:|
| PDE | Partial differential equation |
| FDM | Finite difference method |
| FVM | Finite volume method |
| FEM | Finite element method |
| NURBS | Non-uniform rational B-Splines |
| JFO | Jakobsson–Floberg–Olsson |
| IGA | Isogeometric analysis |
| CAD | Computer-aided design |
| DOFs | Degrees of freedom |
| SPH | Smoothed particle hydrodynamics |

## 2. PHT-Based IGA and the Oil Lubrication Film Model

Although there are many different lubrication models, the typical oil lubrication film considering the classical Reynolds boundary condition in the piston–cylinder system is taken as an example in this article. Additionally, the method in the presented works can be extended to others. Before introducing our refinement method, it is necessary to introduce some relevant fundamentals in this section to help with understanding, which include brief introductions of the refinement method of the PHT spline, the widely used lubrication model of the piston–cylinder system, and the basic steps of the discretization of the Reynolds equation based on IGA. Schematic diagram for cylinder-piston-rod-crank system is shown in Figure 1, and the definitions of parameters are shown in Table 2.

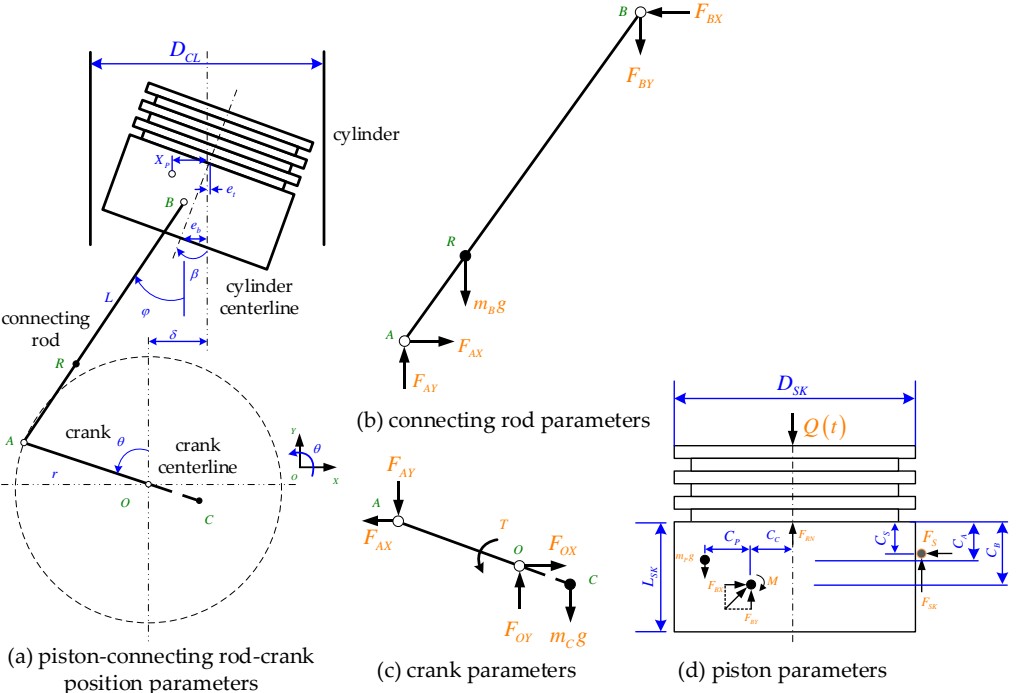

**Figure 1.** Schematic diagram for cylinder–piston–rod–crank system. (**a**) Piston-connecting rod-crank system. (**b**) Parameters of connecting rod. (**c**) Parameters of crank. (**d**) Parameters of piston.

**Table 2.** Parameters and their definitions in piston–cylinder system.

| Parameters | Definitions |
| --- | --- |
| $r$ | Piston skirt radius |
| $R$ | Cylinder liner radius |
| $L_{SK}$ | Piston skirt length |
| $a$ | The long half axis of the ellipse in horizontal cross section of piston |
| $e_t$ | Eccentricity of the upper end of the piston skirt |
| $e_b$ | Eccentricity of the lower end of the piston skirt |
| $d_{et}$ | Derivative of $e_t$ with respect to time |
| $d_{eb}$ | Derivative of $e_b$ with respect to time |
| $C_b$ | Vertical distance from piston pin to top of piston |
| $C_c$ | Distance from piston pin to piston centerline |
| $v$ | Piston reciprocating speed |
| $h$ | Film thickness |

### 2.1. IGA and Local-Refinable PHT Spline

IGA is a new numerical method for solving PDE. Different from FEM, IGA uses the same basis functions as CAD representation rather than the shape functions in meshes approximating to the geometric model in analysis, making it more efficient and accurate.

Actually, there are many splines in graphology that can be used to characterize one geometry model, and all of these spline functions [27] can be used in the IGA.

PHT is a typical local subdividable spline widely used in IGA. It is an example of vertex insertion for local refinement on a PHT [28] mesh. In Figure 2, the blue points represent the boundary vertices, and the green points represent the T nodes, which are generated when refining adjacent elements of different levels. The main work flow of IGA refinement based on PHT spline is shown in Table 3. It should be noted that the T-connection does not change the basis function before the transition to the intersection vertex, and the intersection vertex is represented by the red points. In summary, PHT mesh can be subdivided by inserting crosses in the element, and all meshes have the same degree of calculation accuracy. More details can be found in reference [29].

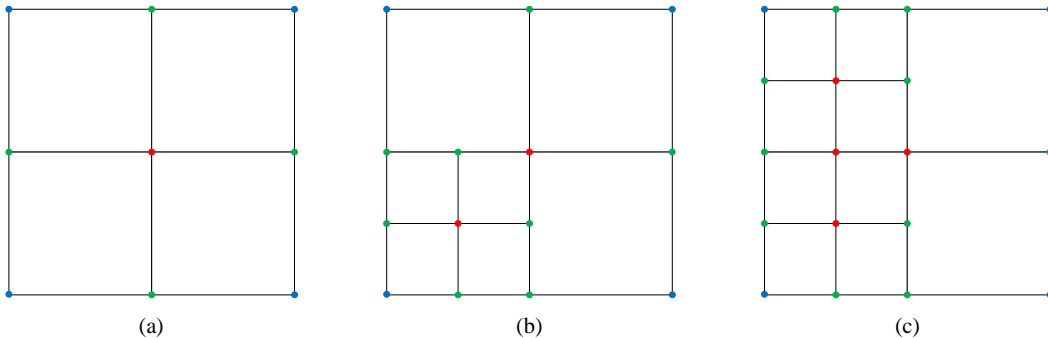

|  (a) | (b) | (c) |

**Figure 2.** Node insertion graph. (**a**) The initial mesh, (**b**) the local subdivision in an element, and (**c**) the local subdivision in the accompanying element.

**Table 3.** PHT-based IGA refinement work flow.

| Steps | Parameters | Remarks |
| --- | --- | --- |
| pretreatment | geometric space | calculate the parameterized $P(u, v)$ of PHT on $\Omega$ |
| solution | parameter space | calculate the results of model on PHT |
| mark | parameter space | mark the target elements |
| recalculate PHT spline | PHT spline | construct PHT spline on subdivided elements |

*2.2. The Reynolds Equation for Lubrication of Piston–Cylinder Interface*

The oil lubrication film between the interfaces of the piston skirt and cylinder forms is shown in Figure 3a, which is always obtained by solving the specific simplified Reynolds equation [30], based on the Navier–Stokes equation and the mass continuity equation.

$$\frac{\partial}{\partial x}\left(\frac{\rho h^3}{12\eta}\frac{\partial p}{\partial x}\right) + \frac{\partial}{\partial y}\left(\frac{\rho h^3}{12\eta}\frac{\partial p}{\partial y}\right) = \frac{\partial}{\partial x}\frac{(u_0 - u_h)\rho h}{2} + \frac{\partial}{\partial y}\frac{(v_0 - v_h)\rho h}{2} + \frac{\partial(\rho h)}{\partial t} \tag{1}$$

In Equation (1), $\rho$ is lubricant density, $p$ is oil film pressure, $\eta$ is dynamic viscosity, $u_0$ and $u_h$ are the film circumferential velocities, respectively, on the piston and cylinder surfaces, $v_0$ and $v_h$ are the axis velocities of oil film, respectively, on the piston and cylinder surfaces, $h$ is film thickness, and $t$ is time.

As shown in Figure 3a, $L_{SK}$ is the length of the piston skirt; $r$ is the crank radius; and $e_t$ and $e_b$ are the distance between the center at the top and bottom of the skirt's center axis and the cylinder's axis. The film lubrication region is always simplified for analysis, as in Figure 3b. The oil lubrication film can be divided to the left lubrication area and the right area, and both of which are symmetrical about plane $\varphi = 0$; so, only the front part is an example. To further simplify Equation (1), the density and viscosity are taken as constants [31], and we assume that there is no leakage of the lubrication.

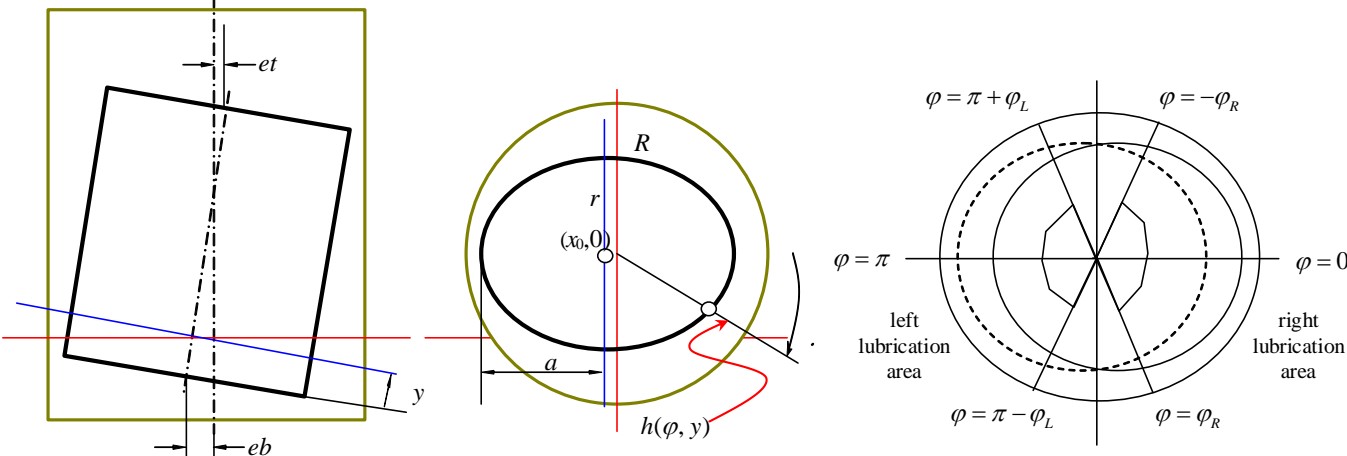

(a) Schematic diagram for piston-cylinder system   (b) Film lubrication region

**Figure 3.** Schematic diagram of cylinder liner–piston system and oil film lubrication area. (**a**) Schematic diagram for piston-cylinder system. (**b**) Film lubrication region.

Take $v_0 = 0, v_h = v$,

$$\frac{\partial}{\partial x}\left(\frac{h^3}{12\eta}\frac{\partial p}{\partial x}\right) + \frac{\partial}{\partial y}\left(\frac{h^3}{12\eta}\frac{\partial p}{\partial y}\right) = -\frac{v}{2}\frac{\partial h}{\partial y} + \frac{\partial h}{\partial t} \tag{2}$$

In order to normalize the domain $D$ into $[0,1] \times [0,1]$, make the following transformations (take the right side as an example):

$$x = R\varphi, \varphi = \varphi_R, y = L_{SK}Y$$
$$D = \{\varphi|0 \le \varphi \le \varphi_R\} \times \{y|0 \le y \le L_{SK}\}, 0 < \varphi_R < \frac{\pi}{2} \tag{3}$$
$$\frac{\partial}{\partial X}\left(-\frac{h^3}{12\eta R^2\varphi_R^2}\frac{\partial p}{\partial X}\right) + \frac{\partial}{\partial Y}\left(-\frac{h^3}{12\eta L_{SK}^2}\frac{\partial p}{\partial Y}\right) = \frac{v}{2}\frac{\partial h}{\partial y} - \frac{\partial h}{\partial t}, (X, Y) \in [0,1] \times [0,1]$$

The film thickness $h$ is the distance between the piston and the cylinder liner at point $(\varphi, y)$ in the simplified lubrication model, which can be calculated through the ellipse method as in Equation (4). Other models considering the microform of the piston skirt and other conditions can be calculated in a similar way, with slight differences in certain parameters, which can be found in [32].

$$h(\varphi, y) = R - \frac{x_0 r^2 cos(\varphi) + ar\sqrt{(a^2 - x_0^2)sin^2(\varphi) + r^2 cos^2(\varphi)}}{\left[a^2 sin^2(\varphi) + r^2 cos^2(\varphi)\right]} \tag{4}$$

where

$$a = \frac{rL_{SK}}{\sqrt{L_{SK}^2 - (e_t - e_b)^2}}, \quad x_0 = e_b + \frac{y}{L_K}(e_t - e_b)$$

In particular, it should be noted that IGA is deficient in dealing with complex geometric models, and is only available for regular-shaped geometric models. The oil lubrication film area is transformed into a rectangular area after normalization for solving, satisfying the requirement of IGA.

### 2.2.1. The Reynolds Boundary Condition

There are several lubrication models for the study of the piston–cylinder system, of which the Reynolds boundary condition is a classic one. It considers that the oil film is

discontinuous, and the rupture at the end of the oil film is a natural phenomenon, as shown in Equation (5).

$$P(\varphi, y)|_{\varphi=\varphi_1} = 0, \ P(\varphi, y)|_{y=0} = 0, \ P(\varphi, y)|_{y=L_{SK}} = 0, \ \left.\frac{\partial P(\varphi, y)}{\partial \varphi}\right|_{\varphi=0} \tag{5}$$

where $\varphi = \varphi_1$ is the boundary of the oil film lubrication area in circumferential direction; $y = 0$ is the bottom boundary and $y = L_{SK}$ is the top boundary; and $\varphi = 0$ is the symmetrical plane. After normalization comes the following:

$$P(X, Y)|_{X=0} = 0, \ P(X, Y)|_{Y=0} = 0, \ P(X, Y)|_{Y=1} = 0, \ \left.\frac{\partial P(X, Y)}{\partial X}\right|_{X=0} \tag{6}$$

### 2.2.2. The JFO Boundary Condition

However, the Reynolds boundary ignores the presence of cavities in the oil lubrication film, which have been confirmed to have a great influence on the lubrication condition. In fact, the lubrication state between the piston cylinder sleeves is very complicated. Cavitation is a complex phenomenon, which usually occurs in the regions where the liquid pressure is below the cavitation pressure. Jakobsson, Flober [26], and Olsson [33] proposed the JFO (Jakobsson–Floberg–Olsson) boundary condition, which is based on the mass conservation of the oil film boundary. JFO theory incorporates the Reynolds boundary condition and provides the oil film reformation boundary condition. The mass conservation model based on the JFO boundary condition [34] is shown as follows:

$$\frac{\partial}{\partial x}\left(\frac{h^3}{12\eta}\frac{\partial p}{\partial x}\right) + \frac{\partial}{\partial y}\left(\frac{h^3}{12\eta}\frac{\partial p}{\partial y}\right) = -\frac{v}{2}\frac{\partial((1-\theta)\rho h)}{\partial y} + \frac{\partial((1-\theta)\rho h)}{\partial t} \tag{7}$$

Additionally, the JFO boundary is

$$\begin{cases} p > 0 \Rightarrow \theta = 0 \\ \theta > 0 \Rightarrow p = 0 \\ \quad 0 \leq \theta \leq 1 \end{cases} \tag{8}$$

where $h$ is the oil film thickness; $\rho$ is the oil film pressure; and $\theta$ is the cavitation factor, affected by the oil film pressure $p$.

From the above equation we can see that the effect of the JFO boundary condition will be directly expressed in the pressure of the oil film. So, the refinement strategy based on the physical features of the pressure value and its variation is still suitable for this lubrication model. Despite different lubrication models considering different states of oil film lubrication, they share the same core which is that the differences in all models are reflected in the distribution of the oil film, and we can capture these features quite clearly through the values of oil film pressure and its gradient. Then, we can conclude that the refinement strategy based on the physical parameters is suitable for different lubrication models.

## 3. IGA Approach for Solving the Reynolds Equation

Before using PHT in solving the Reynolds equation, it is necessary to introduce the discretization of the Reynolds equation in a weak form, which mainly includes the normalization of the general form of the Reynolds equation, the transformation of the solution domain to the element interval to facilitate the solution of the equation, and the assembling of the final stiffness matrix. As the boundary condition makes little difference in the discrete process, the Reynolds boundary condition is used as an example in this section.

Let

$$k_1 = -\frac{h^3}{12\eta R^2 \phi_R^2}, k_2 = -\frac{h^3}{12\eta L_{SK}^2}, f = -\frac{v}{\partial y}\frac{\partial h}{2} + \frac{\partial h}{\partial t} \tag{9}$$

Then, Equation (1) can be converted to

$$\frac{\partial}{\partial x}\left(k_1\frac{\partial p}{\partial x}\right) + \frac{\partial}{\partial y}\left(k_2\frac{\partial p}{\partial y}\right) = f \text{ or } \nabla(k\nabla p) = f \tag{10}$$

Introduce the derivative of the stiffness matrix of the Reynold equation; then, Equation (3) can be transformed into the following form:

$$a(p,v) = -\iint_D \nabla P^T C \nabla v \, dx \, dy \tag{11}$$

If the physical domain $D$ has been transformed via geometric mapping, the transformation method is as follows:

$$F: D_0 \rightarrow D, F(u,v) = \begin{pmatrix} x \\ y \end{pmatrix}$$

Use PHT spline functions to define $F$ in IGA

$$F(u,v) = \sum_{i=1}^m \sum_{j=1}^n R_{ij}(u,v)C_{ij} \tag{12}$$

where $N_{ij}$ is the basis function whose value is 0 on $\Gamma$ and $C_{ij} \in D$

Based on the chain rule of $P(x,y) = P(F(u,v))$, the differentiation form can be written as

$$\nabla_{(x,y)}P(x,y) = DF(u,v)^{-T}\nabla_{(u,v)}P(u,v) \tag{13}$$

where $DF(u,v)$ is a $2 \times 2$ Jacobian matrix.

$$DF(u,v) = \begin{pmatrix} \frac{\partial F_1}{\partial u} & \frac{\partial F_1}{\partial v} \\ \frac{\partial F_2}{\partial u} & \frac{\partial F_2}{\partial v} \end{pmatrix} \tag{14}$$

The approximate solution $p$ can be expressed as the following form through the PHT basis function and the point $q_{ij} \in R$

$$P(u,v) = \sum_{i=1}^m \sum_{j=1}^n R_{ij}(u,v)q_{ij} \tag{15}$$

As the physical domain $D$ has been normalized to $[0,1] \times [0,1]$, the element stiffness matrix $A_e$ can be calculated as follows:

$$
\begin{aligned}
A_e &= \left(\iint_{De} \left(DF(u,v)^{-T}\nabla R_J\right)^T C(DF(u,v)^{-T}\nabla R_I)|\det D\,F(u,v)|du\,dv\right)_{mn \times mn} \\
&= \left(\iint_{De} (\nabla R_J)^T C(\nabla R_I)du\,dv\right)_{mn \times mn}
\end{aligned}
\tag{16}
$$

where

$$C = \begin{pmatrix} k_1\left(h\left(u_p^{(e)}, v_q^{(e)}\right)\right) & 0 \\ 0 & k_2\left(h\left(u_p^{(e)}, v_q^{(e)}\right)\right) \end{pmatrix}$$

Finally, the stiffness matrix $A_e$ can be derived as

$$A = \sum_{e,p,q} w_{e,p,q}\Phi\left(u_p^{(e)}, v_q^{(e)}\right) = \sum_{e,p,q} w_{e,p,q}\left(c_{11,e,p,q}U_{e,p,q}U_{e,p,q}^T + c_{22,e,p,q}V_{e,p,q}V_{e,p,q}^T\right) \tag{17}$$

where $w_{e,p,q}$ is the weight of the Gauss point in element $e$ at $\left(u_p^{(e)}, v_q^{(e)}\right)$, $u_{e,p,q}$ and $v_{e,p,q}$ are the constructive vector quantities, and $c_{11,e,p,q}$ and $c_{22,e,p,q}$ are the coefficients for the calculation of the thickness of the oil film at the point.

Accordingly, the right-hand-side vector $b = (\langle f, R_I \rangle)$ can be calculated as

$$f, R_I = \iint_{D0} (f R_I)(F(u,v)) |detDF(u,v)| dudv, I = 1, 2, \cdots, mn. \tag{18}$$

## 4. Refinement Feature Based on Oil Lubrication Film Pressure Distribution

The final purpose of solving the Reynolds equation is to obtain the pressure distribution of the lubrication film. Thus, the fundamental principal of the strategy in this article is to refine the critical area adaptively signed by the features' extraction from current solving film pressure distribution. Usually, regions with large overall pressure values or large variation in pressure values contribute more to the calculation error related to the physical properties of the oil film. So, the refinement features should be recognized based on the parameters of the pressure values and their variation. For convenience, all of the parameters and their concise meanings used in this chapter are listed in Table 4.

**Table 4.** Parameters and their meanings.

| Parameters | Definitions |
|---|---|
| $T_\theta$ | Cavity indicator |
| $\theta$ | Cavity factor |
| $S_e$ | Element size |
| $L$ | Mesh level |
| $F_e$ | Element type |
| $p_e, p_e'$ | Element pressure value in present mesh and last mesh, respectively |
| $w_{x,y,z}, w_{x,y,z}'$ | Element pressure gradients in different directions in present mesh and last calculation, respectively |
| $D_p$ | Parameter for measuring the magnitude of pressure values |
| $D_{wx,wy,wz}$ | Parameter for measuring the magnitude of pressure gradients |
| $E_p$ | Calculation error of pressure value |
| $E_{ps}$ | Calculation error of average pressure value |
| $E_{wx,wy,wz}$ | Calculation error of pressure gradient |

### 4.1. Refinement Feature Based on Pressure Values

For a certain oil film pressure distribution, the pressure value is the most intuitive parameter. An example of refinement according to the pressure is shown in Figure 4, where the depth of the grayscale and the number represent the magnitude of the pressure value.

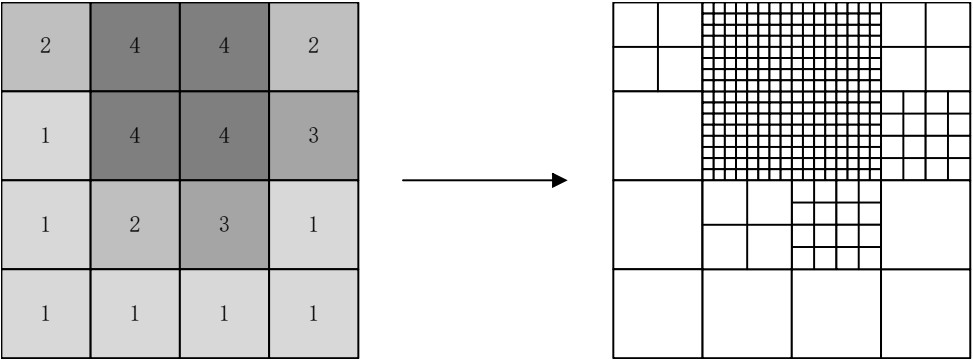

**Figure 4.** A certain pressure distribution of oil film and its expected refinement mesh.

In order to achieve the desired refinement effect, the following parameters were taken into consideration. As mentioned above, the pressure value of each point in the domain

can be calculated as in Equation (15), and then the integration of the pressure value $p_e$ of each element can be calculated as follows:

$$
\begin{aligned}
p_e &= \iint_{D_e} P(u,v)q_{ij}dudv \\
&= \sum_{e,p,q} R_e w_{e,p,q} U_{e,p,q} V_{e,p,q}
\end{aligned}
\tag{19}
$$

Firstly, a base value $p_a$ is introduced here to help assess the magnitude of the pressure value. Then, $D_p$ is defined to characterize the magnitude of the pressure value in the present element.

$$
D_{pm} = \frac{|p_e| - |p_a|}{|p_a|}, D_p = \begin{cases} D_{pm} & (D_{pm} > 0) \\ 0 & (D_{pm} \le 0) \end{cases}
\tag{20}
$$

where

$$
p_a = \lambda p_N' p_N' = \frac{\iint_D P\prime(u,v)q_{ij}dudv}{N} = \frac{\iint_D P\prime(u,v)q_{ij}dudv}{\frac{S}{S_e}}
$$

Here, $p_a$ is the criterion of pressure value in $D_p$. It is suggested as $\lambda p_N'$. $\lambda$ is a constant depending on the accuracy requirement; $p_N'$ is the average pressure value of the element normalized to $S_e$ size.

The accuracy of the calculation result of the pressure value of the element has an impact on the overall result not only in terms of its value, but also in terms of the size of the element. Thus, $S_e$ is introduced to indicate the size of the element. Then, there are many other parameters to be considered, such as $L$, the level of the present mesh (depending on the mesh step), $F_e$, the element type (the level of the element or the size of element), and whether the element is subdivided into the last mesh.

Additionally, a special parameter $T_\theta$ is introduced here to indicate whether the element region is a cavity area or not for the JFO boundary condition. $T_\theta = 0, \theta = 0$ means that the element is in a fully lubricated area; $T_\theta = 1$ means that the element is in the cavity region. $\theta$ is a unique parameter for cavity elements. In order to accelerate the solving speed, the Fischer–Burmeister–Newton–Schur (FBNS) equation [35] is used in this article. Then, the constraint Equation (8) can be modified as follows:

$$
p + \theta - \sqrt{p^2 + \theta^2} = 0
\tag{21}
$$

Since the presence of cavities has a large effect on the distribution of oil film pressure, the cavity region needs to be sufficiently refined to meet the accuracy requirements. Thus, $T_\theta$ is a primary parameter to be considered in the judgement of the refinement process. The parameter $\theta$ can be solved according to reference [36,37], and the details about cavity elements are discussed in Section 5.2.

From another perspective, the calculation error of the pressure values in the results of the two adjacent steps can also be helpful to determine the calculation accuracy of the element. Thus, the calculation results of all of the elements of the last mesh are stored in a vector. If the element has not been subdivided by the last time, then $p_e'$ is the result of the last calculation result of the pressure value, and $S_e'$ is the size of the element; otherwise, $p_e'$ is the pressure value of the parent-level element, and $S_e'$ is the size of the parent-level element. Thus, for the element which has been subdivided by the last time, the error $E_p$ of the pressure values and the error $E_{ps}$ of the average pressure values can be calculated as follows:

$$
E_p = \left| \frac{p_e - p_e'}{p_e'} \right|, E_{ps} = \left| \frac{\frac{p_e}{S} - \frac{p_e'}{S\prime}}{\frac{p_e'}{S\prime}} \right|
\tag{22}
$$

In particular, for the element which has not been subdivided by the last time, the error $E_p$ of the pressure is calculated as follows:

$$E_p = \left| \frac{p_e - k p_e'}{p_e'} \right| \tag{23}$$

where $k$ depends on the type of grid element and the refinement method of the element. It can be seen in Figure 2 that $k = 4$ in PHT meshes.

*4.2. Refinement Feature Based on Variation in Pressure Value*

Meanwhile, regions with rapid variation in pressure are taken into consideration too, as the calculation accuracy of these regions also has a great impact on the final results. Unlike the previous approach with a single scalar value, variation is often in the form of a vector. So, refinement feature evaluating needs to be calculated from multiple directions. For the simplicity of the algorithm and the computational efficiency, the variations in pressure values are calculated as the gradient of pressure in the $x$ direction, $y$ direction, and in the joint direction (represented by $z$), as follows:

$$w_x = \frac{\partial P}{\partial u} = \sum \frac{\partial R}{\partial u} DF(u,v)^{-T} \nabla(u,v) P(u,v) c_{ij} \tag{24}$$

$$w_y = \frac{\partial P}{\partial v} = \sum \frac{\partial R}{\partial v} DF(u,v)^{-T} \nabla(u,v) P(u,v) c_{ij} \tag{25}$$

$$w_z = \frac{\sqrt{\left(\frac{w_x}{X}\right)^2 + \left(\frac{w_y}{Y}\right)^2}}{X^2 + Y^2} \tag{26}$$

$w_a$ is introduced here as the base value, which is suggested to be set according to the results of the last calculation, and $w_{ax}$ is for the $x$ direction; $w_{ay}$ is for the y direction and $w_z$ is for the joint direction. $D_w$ is defined here to measure the degree of the pressure gradient value, and $D_{wx}$, $D_{wy}$, and $D_{wz}$ are the parameters for the $x$ direction, the $y$ direction, and the joint direction, respectively.

$$D_{wx} = \frac{|w_x| - |w_{ax}|}{|w_{ax}|}, D_{wy} = \frac{|w_y| - |w_{ay}|}{|w_{ay}|}, D_{wz} = \frac{|w_z| - |w_{az}|}{|w_{az}|}$$
$$D_{wm} = max(D_{wx}, D_{wy}, D_{wz}), D_w = \begin{cases} D_{wm} & (D_{wm} > 0) \\ 0 & (D_{wm} \leq 0) \end{cases} \tag{27}$$

Accordingly, the calculation errors of the pressure gradient in two adjacent steps are taken to help. Additionally, the error of the pressure gradient in the adjacent steps is shown below:

$$E_{wx} = \left| \frac{w_x}{w_x'} \right|, E_{wy} = \left| \frac{w_y}{w_y'} \right|, E_{wz} = \left| \frac{w_z}{w_z'} \right| \tag{28}$$

where $w_x'$, $w_y'$, and $w_z'$ are the pressure gradient values at the last time calculated in different directions, respectively.

## 5. Adaptive Refinement Strategy

Several parameters are listed based on pressure characteristics in Table 5 to help to judge whether an element should be subdivided or not. From the discussion in the precious section, the final refinement strategy should be developed from two perspectives. The basic principles are proposed here.

**Table 5.** Parameters in refinement strategy.

| Parameters | Type | Definitions |
|---|---|---|
| $p_a$ | Base value | Base value of pressure value |
| $w_a$ | Base value | Base value of pressure gradient |
| $a$ | Hyperparameters | Weight of pressure value error in Principle 1 |
| $b$ | Hyperparameters | Weight of $E_{ps}$ in Principle 1 |
| $d$ | Hyperparameters | Weight of pressure gradient error in Principle 2 |
| $\tau_1, \tau_2$ | Hyperparameters | Judgement threshold in Principle 1 and 2, respectively |
| $\alpha$ | $\tau_1, \tau_2$ uning parameter | Weight of the whole pressure value part |
| $\beta$ | Tuning parameter | Weight of the whole pressure gradient part |
| $\xi$ | Hyperparameters | Judgement threshold in adaptive refinement strategy |

*5.1. Basic Principles*

$$\text{Principle 1}: \ n_1 = D_p + a \cdot (E_p + b \cdot E_{ps}), n_1 > \tau_1 \tag{29}$$

Principle 1 is a judgment criterion for the pressure-based refinement feature. $n_1$ is a composite parameter that measures the magnitude of the element pressure value. $\tau_1$ is proposed as the criterion value. In Equation (29), $a$ is the weight for the whole error part, and $b$ is the weight of $E_{ps}$. $\tau_1$ is the criterion value to be compared with $n_1$; if $n_1 > \tau_1$, it means that the element is of a larger pressure value than the others, and the element should be marked and subdivided in the next step.

$$\text{Principle 2}: \ n_2 = D_w + d \cdot E_w, n_2 > \tau_2 \tag{30}$$

where $D_w = max(D_{wx}, D_{wy}, D_{wz})$, $E_w = max(E_{wx}, E_{wy}, E_{wz})$.

Principle 2 is based on the pressure gradient value referred to as the refinement feature based on the variation in pressure value. Equation (30) shows that $n_2$ is also a composite parameter measuring the magnitude of the pressure variation in the element. $d$ is the weight of the calculation error of the pressure variation. $\tau_2$ is the criterion value to support the judgement; if $n_2 > \tau_2$, which means that the pressure variation in the element is larger than the others, then the element will be marked.

As can be seen above, in Principle 1, elements with a large value will be marked, while elements with a large variation in pressure value will be treated by Principle 2. Additionally, all of the marked elements according to the two principles should be subdivided in the next refinement step. Thus, a final adaptive refinement strategy is proposed to combine the two principles for better refinement results, as shown below.

The final adaptive refinement Principle 3:

$$m = \alpha \cdot \left(D_p + a \cdot \left(E_p + b \cdot E_{ps}\right)\right) + \beta \cdot (D_w + d \cdot E_w), m > \xi \tag{31}$$

where $\alpha$ is the weight of the whole pressure value part, and $\beta$ is the weight of the whole pressure gradient part.

*5.2. Refinement Work Flow*

According to the principles developed above, an adaptive refinement method can be organized. In order to explain the refinement work flow, the concise algorithm is shown in Algorithm 1 and the adaptive refinement strategy is shown in Figure 5.

---

**Algorithm 1**: Refinement strategy algorithm

---

**Input:** Basic parameters of PHT mesh, preliminary solution, and tolerance range
Set the base value $p_a, w_a$ and load the hyperparameter $a, b, d$ for all groups;
For each element:
  Calculate the original information $S_e, L, F_e, p_e, w_{x,y,z}$, and save these results in vector $[dp]$;
  Load the results of the last calculation $p'_e, w'_{x,y,z}$;
  Calculate $D_p, D_w$ and $E_p, E_{ps}, E_w$;
  Classify the element to the specific group and load the base values and hyperpa-rameters of the
corresponding group;
  Check whether the element is in a cavity area;
  Calculation for Principle 1: calculate $n_1$ and compare it with $\tau_1$;
  Calculation for Principle 2: calculate $n_2$ and compare it with $\tau_2$;
  Calculate $m$ and compare it with $\xi$;
  If $m > \xi$
   mark the element;
   If there are elements that fit the condition in Equations (32)–(34)
    update the parameters $\alpha, \beta$.
   End if
  End if
Subdivide the marked elements;
Update the mesh;
End for
**Output:** Subdivided mesh

---

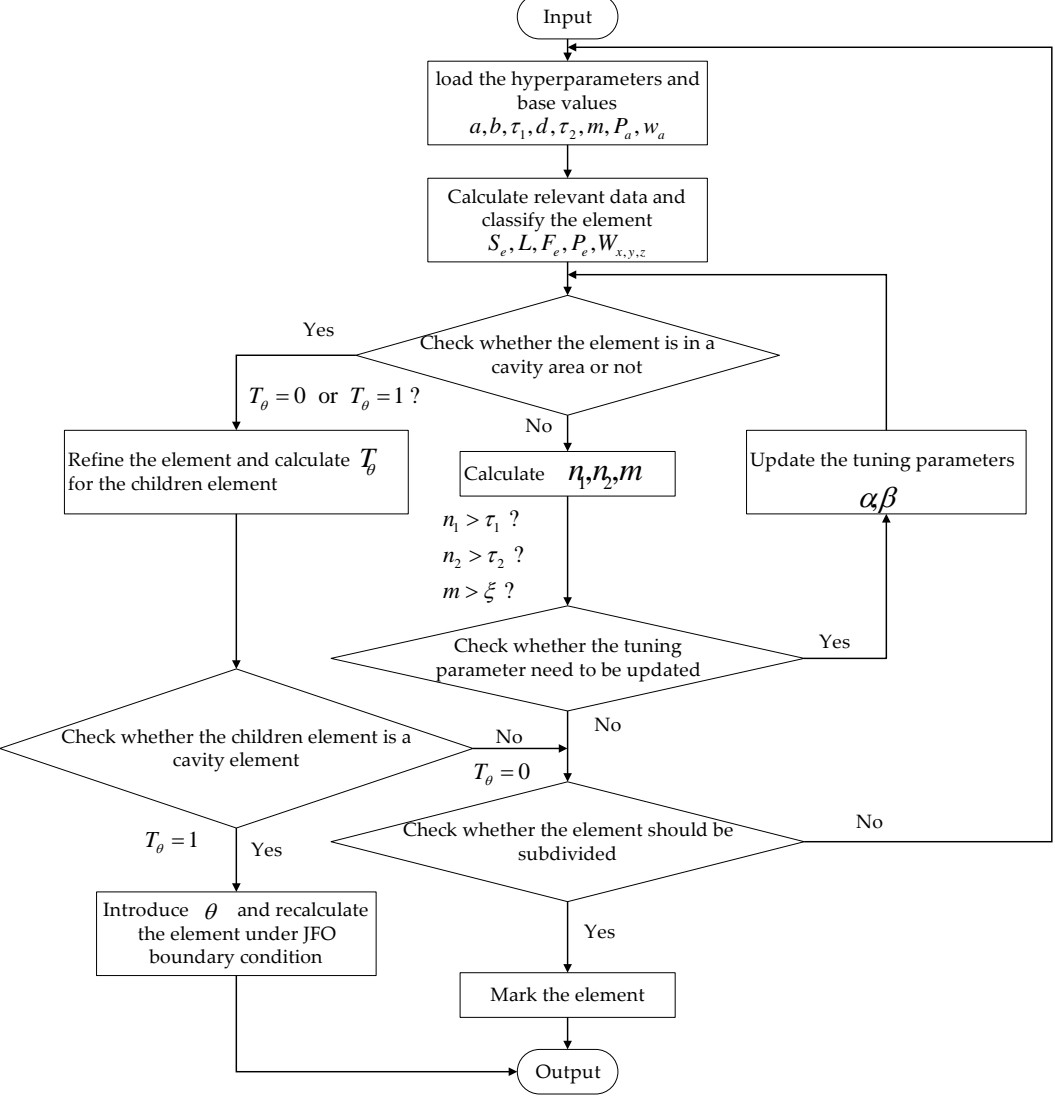

**Figure 5.** Refinement strategy work flow.

In the presented work flow, each element will be classified to a certain group based on its size and degree. It should be noted that the initial parameters are different for different groups. Usually, large elements have a higher weight $\alpha$ of pressure values and small elements have a higher weight $\beta$ of gradient variation.

As for the oil lubrication film problems using the JFO boundary condition, it is necessary to check whether the element is in a cavity region. The calculation of cavity regions is completely different from the fully lubricated area. So, the cavity parameter $T_\theta$ is the primary consideration. Calculate $T_\theta$ and judge whether the element is in a cavity area.

Here are two strategies for the elements in the cavity regions.

(1) The cavity factor $\theta$ is introduced to the marked cavity elements, where $\theta \in 0 \sim 1$. Once an element is marked as a cavity element, it will be calculated as in Equation (21). However, the convergence process will be very inefficient, and it will repeat several times with the mesh updating, which usually leads to numerical oscillations in the solutions due to the convection-dominated terms. Additionally, the cavity formulation based on mortar [37] or other methods [38] would help in this issue.

(2) Once the element is marked as a cavity element, it will be subdivided and its children elements will be recalculated and executed with the same treatment. With the refinement of the mesh, the cavity region could be identified with smaller sizes of elements. Then, the cavity elements will be calculated as in Equation (21). The value of p in the previous calculation can be adopted as the initial value $p_0$ in the iterative process, and then the initial value $\theta_0$ can be calculated too. With the help of these initial values, it will make a great contribution to improving the solving efficiency.

All of the elements will be calculated according to Principle 1, Principle 2, and the adaptive refinement Principle 3. The results of Principle 3 should be consistent with the basic principles as supposed. Otherwise, the tuning parameters $\alpha$ and $\beta$ will be updated automatically, and the specific update method will be introduced in the next section.

### 5.3. Adaptive Update of Parameters

First, parameters $n_1, n_2$, and $m$ must be calculated, respectively. Then, compare $n_1$ with $\tau_1$, $n_2$ with $\tau_2$, and $m$ with $\xi$. If $m > \xi$, the element will be marked and subdivided in the next mesh step. However, there may be conflicts between the adaptive refinement strategy and the two basic principles, shown as follows: If these conditions occur, the parameters $\alpha$ and $\beta$ of the refinement strategy should be updated.

$$\text{Condition 1}: \ n_1 \langle \tau_1, n_2 \rangle \tau_2, m < \xi \tag{32}$$

$$\text{Condition 2}: \ n_1 > \tau_1, n_2 < \tau_2, m < \xi \tag{33}$$

$$\text{Condition 3}: \ n_1 > \tau_1, n_2 > \tau_2, m < \xi \tag{34}$$

Here, Equation (31) is simplified as follows:

$$\alpha \cdot K_1 + \beta \cdot K_2 = 1 \tag{35}$$

where $K_1 = \frac{D_p + a \cdot \left( E_p + b \cdot E_{ps} \right)}{m}, K_2 = \frac{D_w + d \cdot E_w}{m}$.

Conflict Situation 3 can be interpreted as a rule failure for the parameters in Principle 3. For convenience, the elements in Conflict Situation 1 and 3 are classified as the $C1$ element and the elements in Condition 2 are classified as the $C2$ element in this chapter. So, for a certain element class, all of the elements can be divided into subdivided elements, normal elements (no subdivide elements), $C1$ elements, and $C2$ elements in a certain mesh. Then, use points $(K_1, K_2)$ to represent the elements; the refinement strategy can be illustrated as in Figure 6. The solid green line $\alpha x + \beta y = 1$ represents the refinement strategy based on the original parameters, the points of different colors and shapes represent different kinds of elements, and the relationship between the position of the points and the line corresponds to the determination result of the refinement strategy. Therefore, the line

should be shifted down to make most *C*1, *C*2 elements above the line, while the offset distance should be as small as possible. There are three possible situations, as shown in Figure 6. Figure 6a demonstrates that the number of *C*1 elements is far greater than the *C*2 elements. In this situation, $\beta$ is the primary parameter to be updated, and the change in the intercept of the line with the x-axis should be larger than the change in the intercept with the y-axis. The blue dotted line $\alpha\prime x + \beta\prime y = 1$ represents the updated target strategy. Figure 6b represents that the number of *C*2 elements is far greater than the *C*1 elements. Thus, $\alpha$ will be the primary parameter to be updated, accordingly, and the change in the intercept of the line with the y-axis should be larger than the change in the intercept with the x-axis. Figure 6c represents that there is no significant difference in the number of *C*2 elements and *C*2 elements. In this circumstance, the slope of the line will remain the same: $\frac{\alpha}{\beta} = \frac{\alpha\prime}{\beta\prime}$. The parameter updating is transformed into a typical optimization problem, which can be solved using methods such as the least squares method with little computational effort; for the detailed solution procedure, see [39].

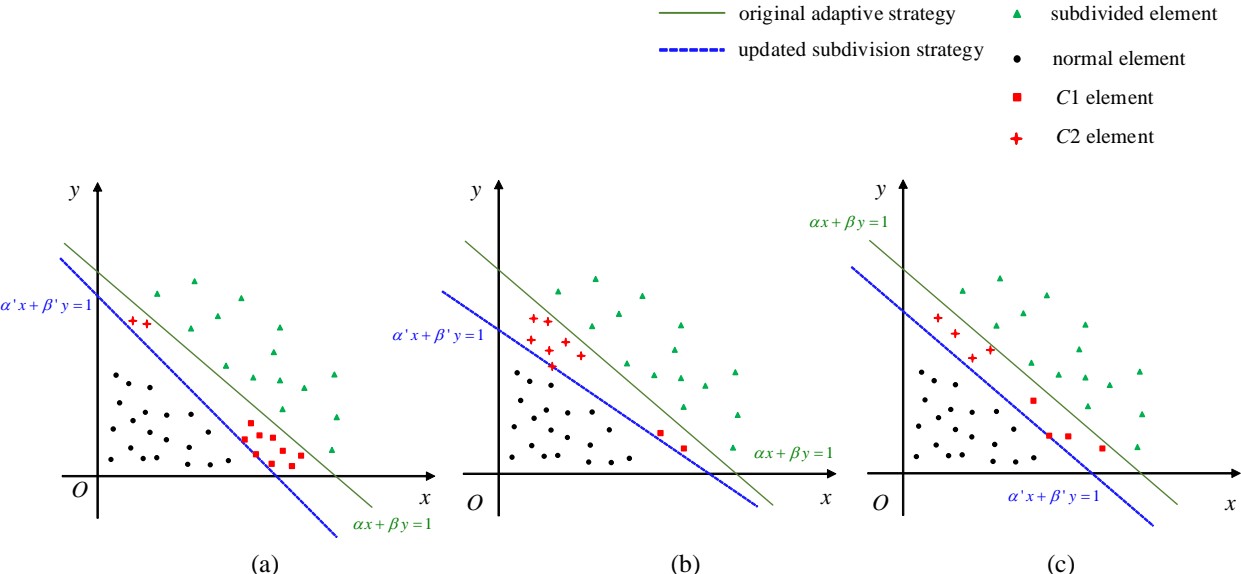

**Figure 6.** Strategy for parameter updates. (**a**) Parameter update strategy for condition 1. (**b**) Parameter update strategy for condition 2. (**c**) Parameter update strategy for condition 3.

## 6. Numerical Examples

In order to verify the rationality of the adaptive refinement strategy, this chapter gives comparisons in dimensions from discretion methods (FEM and IGA), the types of basis functions (NURBS and PHT), the types of subdivision (global equivalent and local refinement), and the subdivision algorithms (residual-based and subdivision feature extraction). Considering the development cycle of these algorithms, only the computational case based on the Reynolds boundary conditions is used here. However, it should be noted that the subdivision strategy in this paper is suitable for the Reynolds equation with the current JFO boundary. As this paper is focused on the solution of the oil lubrication film part of the piston–cylinder system considering the secondary motion of the piston, the calculation of the piston–cylinder system's dynamics section is omitted in this article. The basic input parameters in the numerical examples are from a certain type of single-cylinder gasoline engine, as shown in Table 6 [16].

**Table 6.** Basic input parameters in the calculation example.

| Parameter | Description | Worth | Element |
|---|---|---|---|
| $r$ | Piston skirt radius | $21.725 \times 10^{-3}$ | m |
| $R$ | Cylinder liner radius | $21.75 \times 10^{-3}$ | m |
| $L_{sk}$ | Piston skirt length | $22.5 \times 10^{-3}$ | m |
| $\eta$ | Lubricating oil viscosity | 0.01295 | $Pa \cdot s$ |
| $e_t$ | Eccentricity of the upper end of the piston skirt | $0.78565 \times 10^{-5}$ | m |
| $e_b$ | Eccentricity of the lower end of the piston skirt | $0.24924 \times 10^{-4}$ | m |
| $d_{et}$ | Derivative of $e_t$ with respect to time | $-0.24335 \times 10^{-2}$ | $m \cdot s^{-1}$ |
| $d_{eb}$ | Derivative of $e_b$ with respect to time | $-0.84461 \times 10^{-2}$ | $m \cdot s^{-1}$ |
| $v$ | Piston reciprocating speed | $-10.6543$ | $m \cdot s^{-1}$ |
| $C_b$ | Vertical distance from piston pin to top of piston | $6 \times 10^{-3}$ | m |
| $C_c$ | Distance from piston pin to piston centerline | 0 | m |

Here are some calculation results of the proposed adaptive refinement strategy, as shown in Figure 7. (a) and (b) are the calculation results based on Principle 1 from different perspectives. Figure 7c,d are the calculation results based on Principle 2 from different perspectives. Figure 7e,f are the calculation results based on the adaptive strategy from different perspectives.

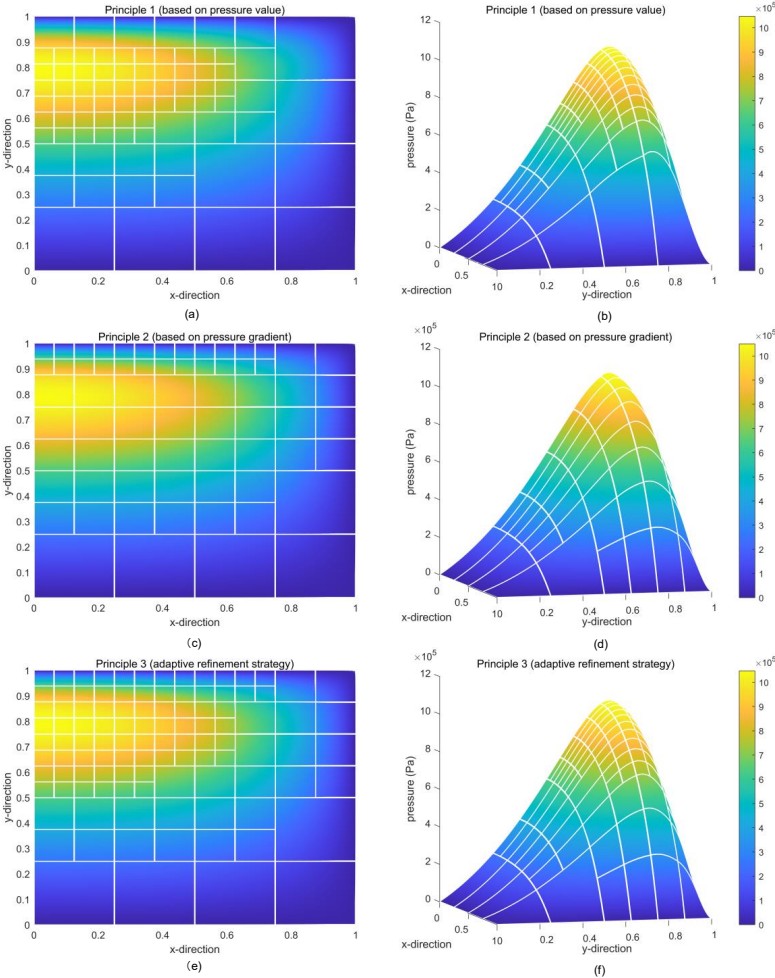

**Figure 7.** Results of the adaptive refinement strategy (the x direction is the circumferential direction, and the y direction is the axial direction). (**a**) Calculation results of Principle 1 in 2D view. (**b**) Calculation results of Principle 1 in 3D view. (**c**) Calculation results of Principle 2 in 2D view. (**d**) Calculation results of Principle 2 in 3D view. (**e**) Calculation results of Principle 3 in 2D view. (**f**) Calculation results of Principle 3 in 3D view.

As can be seen in Figure 7, the sparseness of the grid basically corresponds to the magnitude of the pressure value in Figure 7a,b, whose results are based on Principle 1. The sparseness of the grid basically corresponds to the magnitude of the pressure gradients in Figure 7c,d; they are the results based on Principle 2. However, the elements with large pressure gradient values are ignored in Principle 1, while the elements with large pressure values are dismissed in Principle 2. These areas are also dangerous areas that need to be subdivided. Obviously, if these elements are to be considered, the adaptive refinement Principle 3 is necessary. The elements with large pressure values and pressure gradients are all refined; meanwhile, the meshes are in a coarse size in the uncritical region, as shown in Figure 7e,f.

Table 7 shows the difference in degrees of freedom and convergence between the solutions based on equivalent global refinement strategy and adaptive refinement strategy. The calculation error of the pressure values is based on the calculation results of the densest grid model. The results show that as the refinement proceeds, the pressure value gradually converges to a stable value. The calculation value of the adaptive refinement is already very close to the convergence value when the grid size is much smaller than that of the equivalent global refinement.

**Table 7.** Calculation results based on different refinement strategies.

| Refinement Strategy | | Global Equivalent Refinement | | | Adaptive Refinement | | |
|---|---|---|---|---|---|---|---|
| | | DOFs | Pressure Value (N) | Error (%) | DOFs | Pressure Value (N) | Error (%) |
| Model 1 Refine step | 1 | 36 | −3.8736 | 7.819 | 36 | −3.8736 | 7.819 |
| | 2 | 100 | −3.5914 | 0.036 | 100 | −3.5914 | 0.036 |
| | 3 | 324 | −3.5922 | 0.014 | 236 | −3.5919 | 0.022 |
| | 4 | 1156 | −3.5927 | - | 648 | −3.5923 | 0.011 |
| Model 2 Refine step | 1 | 36 | −23.1833 | 0.620 | 36 | −23.1833 | 0.620 |
| | 2 | 100 | −23.0347 | 0.025 | 100 | −23.0347 | 0.025 |
| | 3 | 324 | −23.0389 | 0.007 | 220 | −23.0396 | 0.003 |
| | 4 | 1156 | −23.0404 | - | 648 | −23.0403 | <0.001 |
| Model 3 Refine step | 1 | 36 | −9.4180 | 0.772 | 36 | −9.4180 | 0.772 |
| | 2 | 100 | −9.4827 | 0.091 | 100 | −9.4827 | 0.091 |
| | 3 | 324 | −9.4938 | 0.026 | 288 | −9.4937 | 0.025 |
| | 4 | 1156 | −9.4913 | - | 980 | −9.4913 | <0.001 |
| Model 4 Refine step | 1 | 36 | −6.1139 | 1.124 | 36 | −6.1139 | 1.124 |
| | 2 | 100 | −6.1706 | 0.207 | 100 | −6.1706 | 0.207 |
| | 3 | 324 | −6.1823 | 0.018 | 272 | −6.1827 | 0.011 |
| | 4 | 1156 | −6.1834 | - | 752 | −6.1832 | 0.003 |

To verify the efficiency improvement in the feature-based adaptive refinement strategy, a numerical experiment of 100 sets of data is given here to compare the solution efficiency. A bubble-function-based [40] error estimation refinement method (one kind of residual-based posteriori error estimation method) was used. The global equivalent subdivision method was used for the calculation of the baseline results, and for the basic control of the calculation results; all of the meshes were refined four times in the solution procession. To show the results of the efficiency comparison more clearly, $t$ represents the solution time, $N$ represents the degrees of freedom, and $\delta$ is the absolute value of the deviation between the calculation result and the baseline value. The results of degrees of freedom and solution time are shown in Figure 8; $\delta/N$ and $\delta/t$ are shown in Figure 9. The red crosses are below the green circles in Figure 9, which means that the results of the feature extraction refinement method carry out a lower deviation per degree of freedom or calculation time. In other words, the calculation efficiency of the feature extraction refinement method is better than the bubble-function-based refinement method. And the overall reduction in the

solution time is 72.24% compared to the equivalent refinement method, and it is 10.86% for the bubble-function-based adaptive refinement method.

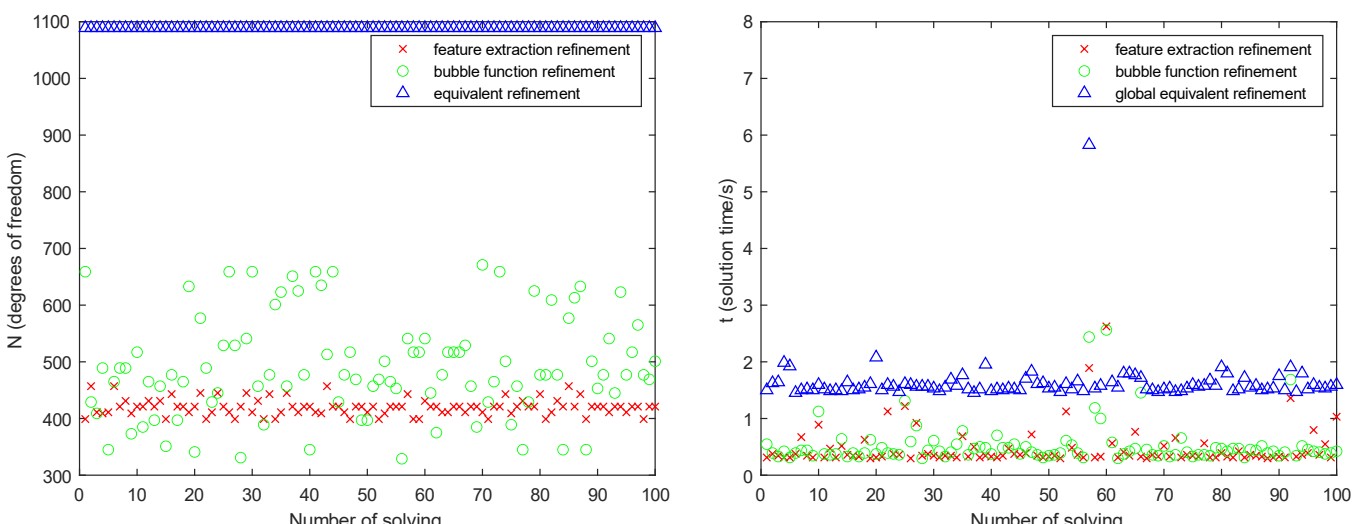

**Figure 8.** Degrees of freedom and solution time.

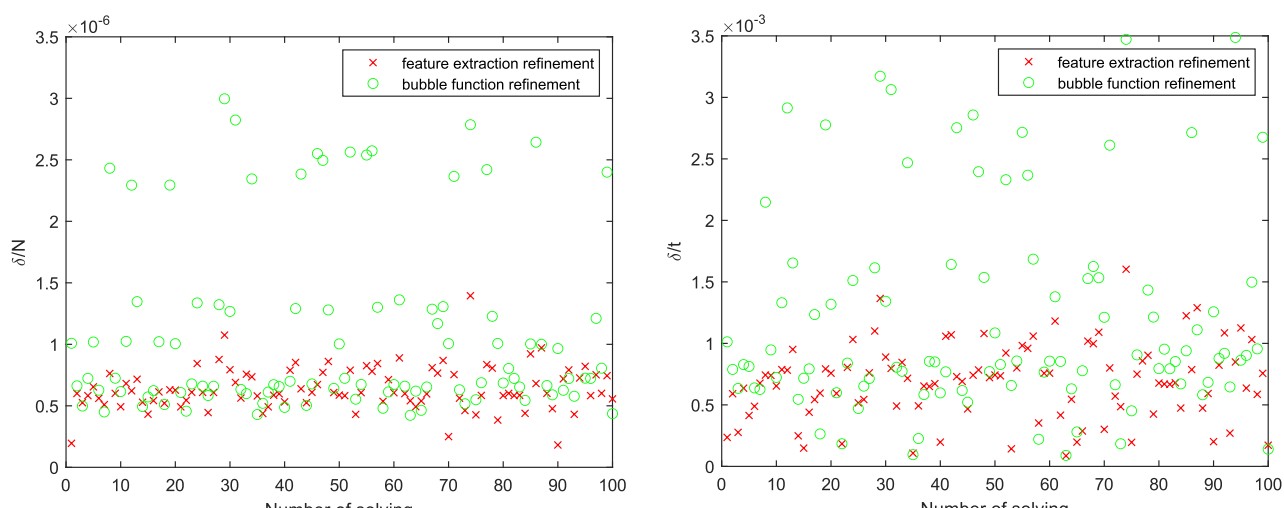

**Figure 9.** Results of $\delta/N$ and $\delta/t$.

In the above discussion, it has been confirmed that the refinement effect based on the feature-extraction-based adaptive refinement strategy basically achieves the target. However, the fundamental aim of mesh refinement work is to improve the efficiency of the solution while maintaining high accuracy; thus, the comparison of source consuming and solving accuracy is shown below. The calculation results of FEM are taken as an example, shown in Figure 10a,b, and the model is calculated using the free triangle grid in Models 1 and 2 and the free quadrilateral grid in Models 3 and 4. The results of IGA obtained using the NURBS spline for calculation based on the equivalent global refinement strategy are shown in Figure 10c,d, and the results of the PHT-based IGA approach based on the adaptive refinement strategy proposed in this paper are shown in Figure 10e,f. The pressure integral value on the domain region is taken as the judgement of solution accuracy, and the DOFs are taken as the consuming computing source, as shown in Figure 11.

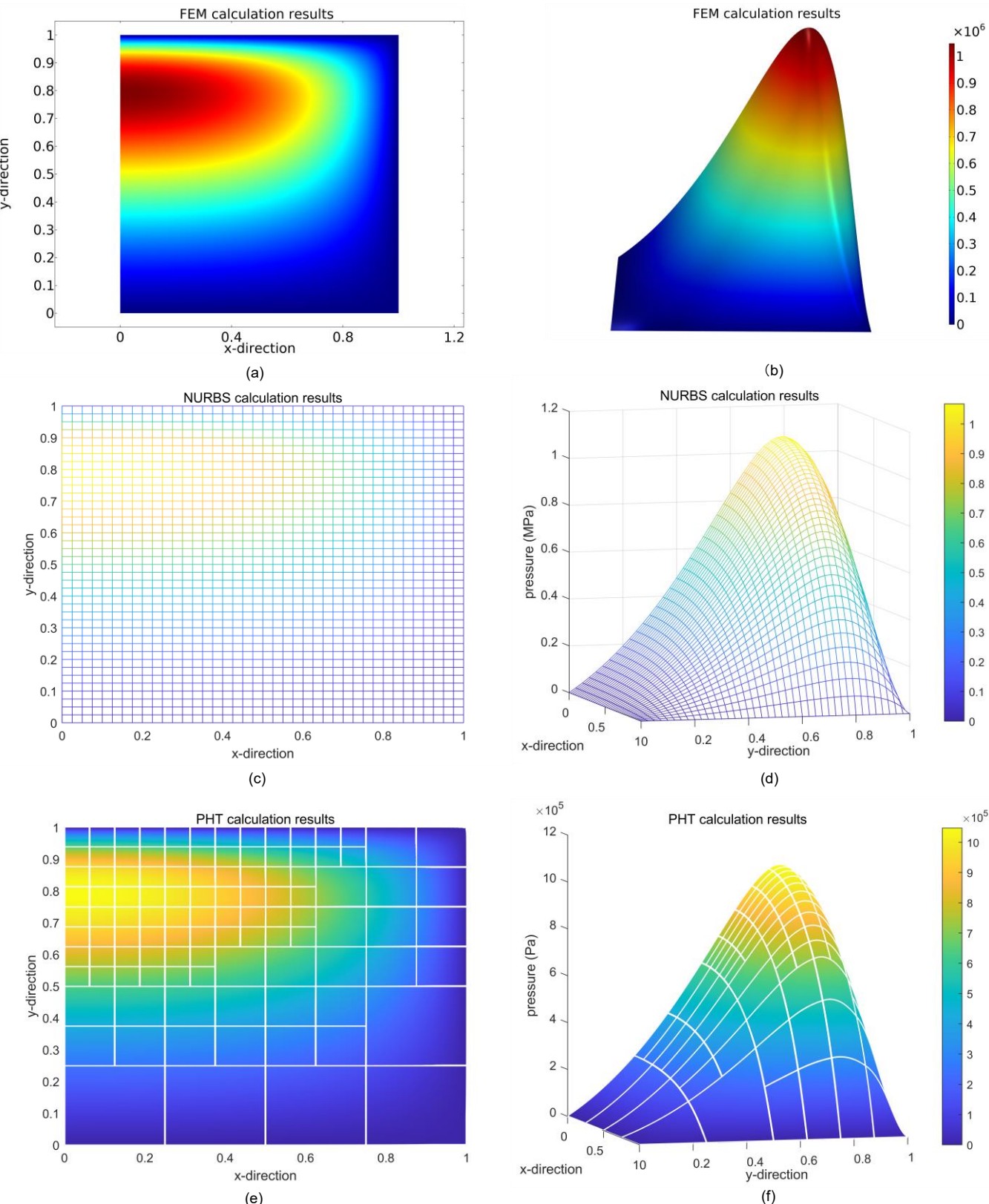

**Figure 10.** Calculation results of FEM, NURBS spline, and PHT spline based on equivalent global refinement. (**a**) FEM calculation results in 2D view. (**b**) FEM calculation results in 3D view. (**c**) IGA calculation results based on NURBS spline in 2D view. (**d**) IGA calculation results based on NURBS spline in 3D view. (**e**) IGA calculation results based on PHT spline in 2D view. (**f**) IGA calculation results based on PHT spline in 3D view.

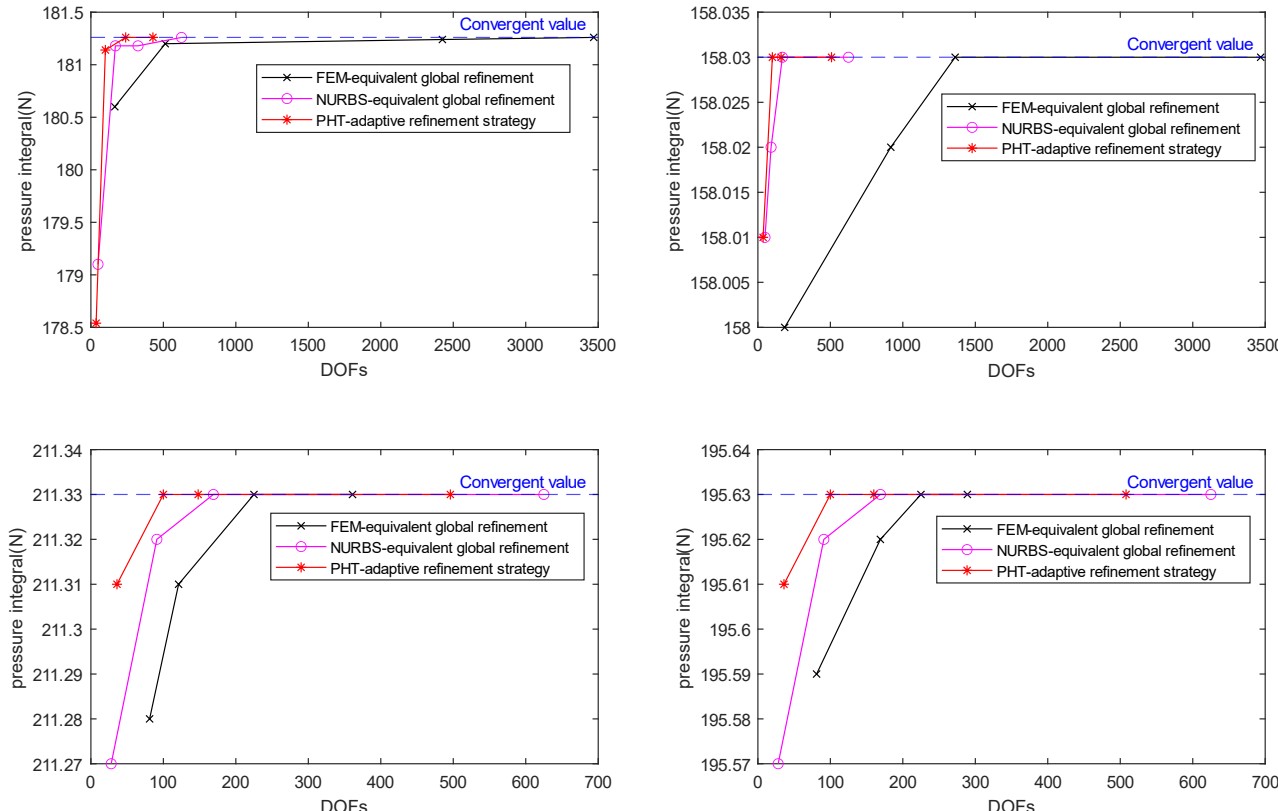

**Figure 11.** Comparison of DOFs and pressure integral.

According to Table 8 and Figure 11, as the degrees of freedom increase, the two IGA algorithms of NURBS and PHT converge faster to the exact value. The convergence results of the PHT approach were obtained at a lower degree of freedom and under less resource consumption. This confirms that the PHT-based IGA approach based on the adaptive refinement strategy performs better both in calculation accuracy and resource consumption.

**Table 8.** Pressure integral and DOF results of piston–cylinder model.

| | Method | Mesh 1 | | Mesh 2 | | Mesh 3 | | Mesh 4 | |
|---|---|---|---|---|---|---|---|---|---|
| | | DOFs | Pressure | DOFs | Pressure | DOFs | Pressure | DOFs | Pressure |
| 1 | FEM | 165 | 180.60 | 513 | 181.20 | 2425 | 181.24 | 3469 | 181.26 |
| | NURBS | 49 | 179.10 | 169 | 181.18 | 325 | 181.18 | 625 | 181.26 |
| | PHT | 36 | 178.54 | 100 | 181.14 | 240 | 181.26 | 428 | 181.26 |
| 2 | FEM | 185 | 158.00 | 917 | 158.02 | 1361 | 158.03 | 3469 | 158.03 |
| | NURBS | 49 | 158.01 | 91 | 158.02 | 169 | 158.03 | 625 | 158.03 |
| | PHT | 36 | 158.01 | 100 | 158.03 | 160 | 158.03 | 508 | 158.03 |
| 3 | FEM | 81 | 211.28 | 121 | 211.31 | 225 | 211.33 | 361 | 211.33 |
| | NURBS | 28 | 211.27 | 91 | 211.32 | 169 | 211.33 | 625 | 211.33 |
| | PHT | 36 | 211.31 | 100 | 211.33 | 148 | 211.33 | 496 | 211.33 |
| 4 | FEM | 81 | 195.59 | 169 | 195.62 | 225 | 195.63 | 289 | 195.63 |
| | NURBS | 28 | 195.57 | 91 | 195.62 | 169 | 195.63 | 625 | 195.63 |
| | PHT | 36 | 195.61 | 100 | 195.63 | 160 | 195.63 | 508 | 195.63 |

## 7. Conclusions

To accelerate the solving speed of the lubrication film in the piston–cylinder system considering the secondary motion of the piston, a PHT-based IGA refinement method based on the feature extraction of the lubrication film pressure was proposed in this article.

With the help of IGA, the approach proposed has a simple calculation process with less computational consumption than others, but the calculation accuracy can be maintained.

In this paper, the local refinement process of PDEs with variable coefficients shaped like the Reynolds equation was implemented by means of feature extraction. The adaptive refinement process marks the elements to be subdivided by identifying the physical parameters of the oil film strongly associated with calculated values such as the element pressure value and the corresponding gradient. This approach avoided the large number of complex calculations required to identify the residuals or to calculate the error characteristics and directly avoided the obstacles posed by the variable coefficients to the identification of the refinement characteristics, and simply used the oil film distribution obtained from the current hierarchical mesh calculation, thus improving the computational efficiency, which saved more than 70% of the solution time compared with the global equivalent subdivision method used in numerical examples. At the same time, for other similar Reynolds equations, the different physical parameters can also be proposed as refinement features to achieve mesh adaptive refinement. Additionally, any local refinement-supported splines can be used in IGA for the implementation of adaptive refinement. Moreover, the adaptive strategy is applicable to different lubrication models, such as the widely used JFO lubrication models.

Of course, later in the feature recognition, deep learning is expected to identify key areas in a global way, which can further improve the recognition efficiency and help to improve the computational efficiency in the system simulation.

**Author Contributions:** Conceptualization, B.Z.; methodology, J.Y. and W.X.; software J.Y. and B.Z.; validation, B.Z. and J.Y.; formal analysis, J.Y. and H.L.; investigation, J.Y. and W.X.; writing-original draft preparation, J.Y. and W.X.; writing-review and editing, J.Y. and B.Z.; supervision, H.L.; project administration B.Z. All authors have read and agreed to the published version of the manuscript.

**Funding:** This research was supported by the Natural Science Foundation of Hubei Province (CN) (Grant No. 2019CFB693), the Research Foundation of the Education Department of Hubei province (CN) (Grant No. B2019003) and the open Foundation of the Key Laboratory of Metallurgical Equipment and Control of Education Ministry (CN) (Grant No. 2015B14). The support is gratefully acknowledged.

**Data Availability Statement:** The data presented in this study are available on request from the corresponding author.

**Conflicts of Interest:** The authors declare no conflict of interest.

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
