# Peer review of "A Feature-Extraction-Based Adaptive Refinement Method for Solving the Reynolds Equation in Piston–Cylinder System"

_lubricants, doi:10.3390/lubricants11030128_

Round 1

Reviewer 1 Report

The paper presents an interesting discussion on the refinement strategies for FEM models used in solving lubrication models. A refinement method based on the feature extraction of the lubrication film pressure is proposed in order to accelerate the solving speed of the model. A comparison with other two strategies is proving the efficiency of the proposed method.

There are very few minor corrections:

Abstract  - Stokes not stock

Eqs. 3 and 9 – The * sign should be eliminated

Author Response

Thank you so much for your review and suggestions. Please see the attachment.

Reviewer 2 Report

Please address the following comments before reconsidering this paper for publication:

Title

The phrase “for IGA solving film lubrication model” does not make sense. Please revise the title.

Abstract

·         Please provide some quantitative demonstration of your refinement approach in the abstract. For example, calculation time savings in % versus a reference uniform solution. What have you achieved by your proposed method which was not obtained by previous studies? This needs a brief mention in the abstract.

·         In general, the abstract can/must be which can b improved from the English and readability standpoints. For example: “For a partial differential equation (PDE) simplified 11 from the Navier-stock equation and the continuity equation, many numerical algorithms such as 12 FDM, FVM, and FEM are available for the solving” needs to be rephrased.

Introduction

·         Due to the excessive number of acronyms used throughout the manuscript, a list of abbreviations must be added for clear referencing.

·         A literature survey shows that, in addition to the family of mesh-based methods reviewed by the authors, the family of mesh-free methods like vortex or Lagrangian particle methods have shown success in solving the Reynolds equation and thin film flows. The authors are required to include at least the following three citations in their literature review:

https://doi.org/10.1145/3450626.3459864

https://doi.org/10.1016/j.cma.2019.112639

https://doi.org/10.1016/j.cma.2018.03.021

and discuss the advantages/disadvantages of such techniques compared to the classical mesh-based schemes:

·         Discussion around previous works, their outcomes, and main limitations, is rather poor. The authors have repeatedly cited a reference without properly describing the issue or shortcoming of that work, which might have led to the proposed framework. Therefore, more critical review of these references (e.g., [4-12]) is needed in order for the reader to be able to draw a conclusion from the state of the art to the present study.

·         The phrase “further improve the solving efficiency” in the last paragraph of Section is not scientifically precise. Please describe the specifics and describe the reason of making this choice.

PHT-based IGA and the oil lubrication film model

·         Please add a new figure before Section 2.1 to illustrate the piston-cylinder system and specify the basic items/definitions.

IGA approach for solving Reynolds equation

·         The derivation procedure and theoretical background of IGS is quite lengthy, while it is not the key novelty of this paper. Please reduce the length of this section by omitting unnecessary expressions and focusing on the main direction of your study, i.e., the refinement algorithm and its offered computational efficiency.

Refinement feature based on oil lubrication film pressure distribution

·         Please describe with more details why “pressure” is the most intuitive value by naming a few alternative parameters that could be chosen instead.

·         In Figure 3, it seems that only 5 resolutions exist in the grid while coloring pattern (i.e., pressure values) does not match this number. Please clarify or choose a clearer illustration style.

Adaptive refinement strategy

·         Before applying the framework to final examples, the authors must provide a preliminary test where they show the convergence of their adaptive IGA-based approach to a reference non-adaptive solution. A self-designed synthetic benchmark analysis would suffice for this verification purpose.

Numerical examples

·         Provide the reference from which you took the material properties in Table 5.

·         The quality of Figure 6 is poor and needs to be improved. Captions and axis labels are too small. More importantly, you need to include the “low” and “high” resolution solutions as the upper and lower bounds for a clear comparison.  

Conclusion

·         Please summarize the key findings of your study in a quantitative form (% in #mesh reduction or % loss of accuracy, etc.)

General Remarks

·         The English language of this paper in its current form is OK in most places. Nonetheless, there are still some punctuation issues, missing articles, wrong structures, and grammatical errors. Please have a native English speaker proofread your text or use a professional language editor.

Author Response

(The authors gave the same response as above.)

Reviewer 3 Report

This paper proposes a novel approach to apply a PHT-based adaptive refinement strategy in a piston-cylinder model, using the Reynolds equation to describe the lubricant flow. It is well-structured and written. It starts with the physical model and different refinement strategies and then compares them with one example.

While using PHT splines is innovative, my main concern is combining the refinement strategy with the JFO boundary condition. The JFO boundary condition is mass-conserved and introduces the new variable theta into the Reynolds equation. It usually leads to numerical oscillations in the solutions due to the convection-dominated terms. However, these issues are not mentioned in the paper. I am curious about how they are handled here.

If occuring, usually, such oscillations can be reduced or eliminated using proper stabilization methods such as SUPG or GLS. Special algorithms based on mortar methods could also accelerate the convergence. I would like to comment that it would be interesting to see how these effects could be implemented combined with the refinement strategy.

In these concerns, the paper would also in general benefit from including some more relevant literature in the introduction:

https://doi.org/10.1016/0045-7825(82)90071-8

https://doi.org/10.1016/j.triboint.2013.12.012

https://doi.org/10.1016/j.cma.2022.115263

Unfortunately, in all plots only the pressure and its convergence is shown, it would also be very interesting to see the distribution of the void fraction.

So, it is unclear which cavitation boundary condition was used in the numerical example presented in Section 6. If the Reynolds boundary condition was used, the paper should include an example applying the JFO boundary condition to verify the refinement strategy shown in Figure 4. If the JFO boundary condition was used, maybe expanding the fluid domain would enable a clearer observation of the cavitated effects. I emphasise this because it is notable that in the chosen example there is pressure in the entire area and the cavitation area thus appears to be very small. 

Author Response

(The authors gave the same response as above.)

Round 2

Reviewer 2 Report

The major issues reported by this reviewer have been addressed in the revised version. The paper can be accepted for publication.

Author Response

Thank you so much for your review. Please see the attachment.

Reviewer 3 Report

Thank you for your response and addressing my concerns. I appreciate the thorough explanation of the two methods used for handling the JFO boundary condition in the future work. However, I noticed that you still did not explicitly mention which cavitation boundary condition (i.e., Reynolds boundary condition) was used in chapter 6. I kindly ask you to include this information in the paper to avoid any confusion for the readers.

Author Response

(The authors gave the same response as above.)
